# Delivery of endocytosed proteins to the cell–division plane requires change of pathway from recycling to secretion

Sandra Richter[1], Marika Kientz[1], Sabine Brumm[1], Mads Eggert Nielsen[1†], Misoon Park[1], Richard Gavidia[1], Cornelia Krause[1], Ute Voss[1‡], Hauke Beckmann[1], Ulrike Mayer[2], York-Dieter Stierhof[2], Gerd Jürgens[1]*

[1]Department of Developmental Genetics, The Center for Plant Molecular Biology (ZMBP), University of Tübingen, Tübingen, Germany; [2]Microscopy, The Center for Plant Molecular Biology (ZMBP), University of Tübingen, Tübingen, Germany

*For correspondence: gerd.
juergens@zmbp.uni-tuebingen.de

Present address: †Department of Plant and Environmental Sciences, University of Copenhagen, Copenhagen, Denmark; ‡Plant Sciences Division, School of Biosciences, University of Nottingham, Nottingham, United Kingdom

Competing interests: The authors declare that no competing interests exist.

**Abstract** Membrane trafficking is essential to fundamental processes in eukaryotic life, including cell growth and division. In plant cytokinesis, post-Golgi trafficking mediates a massive flow of vesicles that form the partitioning membrane but its regulation remains poorly understood. Here, we identify functionally redundant Arabidopsis ARF guanine-nucleotide exchange factors (ARF-GEFs) BIG1–BIG4 as regulators of post-Golgi trafficking, mediating late secretion from the trans-Golgi network but not recycling of endocytosed proteins to the plasma membrane, although the TGN also functions as an early endosome in plants. In contrast, BIG1-4 are absolutely required for trafficking of both endocytosed and newly synthesized proteins to the cell–division plane during cytokinesis, counteracting recycling to the plasma membrane. This change from recycling to secretory trafficking pathway mediated by ARF-GEFs confers specificity of cargo delivery to the division plane and might thus ensure that the partitioning membrane is completed on time in the absence of a cytokinesis-interphase checkpoint.

## Introduction

In post-Golgi membrane trafficking, cargo proteins are dynamically distributed between trans-Golgi network (TGN), various endosomes, lysosome/vacuole and plasma membrane (*Surpin and Raikhel, 2004*). In contrast to animals, the TGN also functions as an early endosome in plants and is a major trafficking hub where secretory, endocytic, recycling and vacuolar pathways intersect (*Viotti et al., 2010*; *Reyes et al., 2011*). Therefore, it has been notoriously difficult to functionally delineate the recycling vs secretory pathways in plants. Sorting of cargo proteins occurs during the formation of transport vesicles, involving activation of small ARF GTPases by ARF guanine-nucleotide exchange factors (ARF-GEFs) and recruitment of specific coat proteins (*Casanova, 2007*). Arabidopsis ARF-GEFs are related to human large ARF-GEFs, GBF1 or BIG1. Whereas the three GBF1-related members GNOM, GNL1 and GNL2 have been characterised in detail (*Geldner et al., 2003*; *Richter et al., 2007, 2012*), of the 5 BIG1-related ARF-GEFs only BIG5 has been analysed so far and implicated in pathogen response (MIN7) and endocytic traffic (BEN1) (*Nomura et al., 2006, 2011*; *Tanaka et al., 2009*; *Tanaka et al., 2013*). Here, we show that ARF-GEFs BIG1-4 play a crucial role in post-Golgi traffic, which enables us to dissect the regulation of secretory and recycling pathways in interphase and cytokinesis.

**eLife digest** Cells are surrounded by a plasma membrane, and when a cell divides to create two new cells, it must grow a new membrane to keep the two new cells apart. Animal cells and plant cells tackle this challenge in different ways: in animal cells the new membrane grows inwards from the surface of the cell, whereas the new membrane grows outwards from the centre of the cell in plant cells.

The materials needed to make the plasma membrane are delivered in packages called vesicles: most of these materials arrive from a structure within the cell called the trans-Golgi network, but some materials are recycled from the existing plasma membrane. In plants the formation of the new cell membrane is orchestrated by scaffold-like structure that forms in the plant cell called the 'phragmoplast'. It is widely thought that this structure guides the vesicles bringing materials from the trans-Golgi network, but the details of this process are not fully understood.

Now, Richter et al. have discovered four proteins, called BIG1 to BIG4, that control the formation of the new cell membrane in the flowering plant *Arabidopsis thaliana*, a species that is routinely studied by plant biologists. These four proteins belong to a larger family of proteins that control the trafficking of vesicles within a cell. Richter et al show that a plant cell can lose up to three of these four proteins and still divide, as the plant can still grow and develop as normal. Thus, BIG1 to BIG4 appear to perform essentially the same role in the plant.

Richter et al. also show that, when a plant cell is not dividing, these proteins are involved in controlling the delivery of new materials to surface membrane, and not the recycling of material. However, when a cell is dividing, these proteins switch to regulate both processes, but direct all the material to a new destination—the newly forming membrane, instead of the established surface membrane. Richter et al. suggest that this switch is important to stop any recycling to the plasma membrane that might move material away from the new membrane. The next challenge will be to identify the molecular signals and mechanisms that enable the proteins BIG1 to BIG4 to re-route the recycling of membrane material during cell division.

## Results

### ARF-GEFs BIG1 to BIG4 are redundantly required in development

Up to three of ARF-GEFs BIG1 to BIG4 (BIG1-4) were knocked out without recognisable phenotypic effect except for *big1,2,3*, which was retarded in growth because BIG4 is predominantly expressed in root and pollen (*Figure 1A*, *Figure 1—figure supplement 1A*). Other triple mutants were growth-retarded only if the activity of the respective fourth gene was reduced to 50%. No quadruple mutants were recovered because BIG1-4 were essential in male reproduction, sustaining pollen tube growth (*Figure 1B*, *Figure 1—figure supplement 1B*). BIG1-4 functional redundancy would be consistent with the occurrence of *BIG1-4*-like single-copy or closely related sister genes in lower plants (*Figure 1—figure supplement 1C*). Although large ARF-GEFs are often inhibited by the fungal toxin brefeldin A (BFA), the SEC7 domain of BIG3 (At1g01960; formerly named BIG2 in *Nielsen et al., 2006*; see nomenclature used by *Cox et al., 2004*) displayed BFA-insensitive GDP/GTP exchange activity in vitro (*Nielsen et al., 2006*). BFA treatment of *big3* mutants impaired seed germination and seedling root growth, in contrast to wild-type (*Figure 1D,E*). We engineered a BFA-resistant variant of the naturally BFA-sensitive ARF-GEF BIG4 by replacing amino acid residue methionine at position 695 with leucine, as previously described for the recycling ARF-GEF GNOM (*Geldner et al., 2003*). Engineered BFA-resistant BIG4-YFP rescued BFA-inhibited seed germination of *big3* (*Figure 1F*). The rescue activity of BFA-resistant BIG4 was comparable to that of BIG3 when both were expressed from the ubiquitin 10 (*UBQ10*) promoter whereas BFA-sensitive BIG4 did not at all rescue BFA-inhibited primary root growth of *big3* mutant seedlings (*Figure 1—figure supplement 1D,E*). Thus, BFA treatment of *big3* single mutants effectively causes conditional inactivation of BIG1-4 ARF-GEF function, providing us with a unique tool for studying BIG1-4-dependent trafficking in an organismic context.

### BIG1 to BIG4 regulate membrane trafficking at the TGN

BIG4-YFP co-localized with TGN markers vacuolar H⁺-ATPase (VHA) subunit a1 and ARF1 GTPase (*Figure 1I–L*, *Figure 1—figure supplement 2O–R*; *Dettmer et al., 2006*; *Stierhof and El Kasmi,*

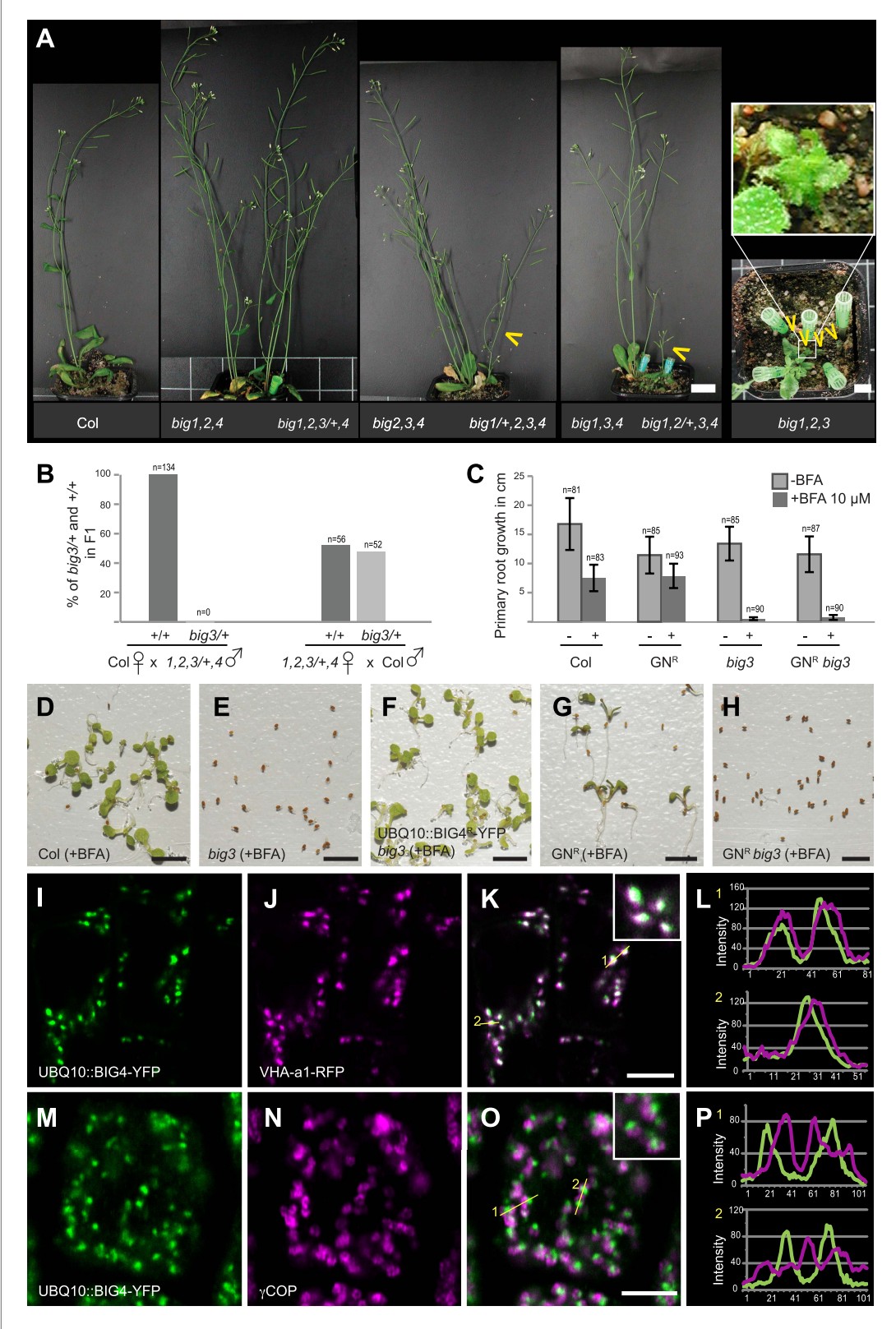

**Figure 1**. BIG1 – BIG4 act redundantly at TGN and are involved in several physiological processes. (**A**) *big1,2,4* (*big1 big2 big4*), *big2,3,4* (*big2 big3 big4*), *big1,3,4* (*big1 big3 big4*) and *big1,2,3/+,4* (*big1 big2 big3/BIG3 big4*) mutant plants without obvious phenotype but *big1/+,2,3,4* (*big1/BIG1 big2*

*Figure 1. Continued on next page*

*Figure 1. Continued*

*big3 big4*), *big1,2/+,3,4* (*big1 big2/BIG2 big3 big4*) and *big1,2,3* (*big1 big2 big3*) were dwarfed (yellow arrowheads). Scale bar, 2 cm. (**B**) F1 of reciprocal crosses between wild-type (Col) and *big1 big2 big3/BIG3 big4* (*1,2,3/+,4*) mutants: 0% or 48% *big3* heterozygous seedlings derived from mutant male or female gamete, respectively. (**C**) BFA inhibited primary root growth of *big3* mutant seedlings with or without BFA-resistant GNOM (GN$^R$ *big3*). Numbers of analysed seedlings are indicated (**B** and **C**). (**D-H**) BFA treatment did not prevent seed germination in wild-type (Col; **D**) and BFA-resistant GN (GN$^R$; **G**) but did so in *big3* mutants without (**E**) or with BFA-resistant GNOM (GN$^R$ *big3*; **H**). This defect was suppressed by BFA-resistant BIG4 (UBQ10::BIG4R-YFP *big3*; **F**). Scale bar, 5 mm. (**I-L**) Live imaging of BIG4-YFP (**I**) and TGN marker VHA-a1-RFP (**J**) revealed co- localization (**K**; **L**, intensity–line profile). (**M–P**) Immunolocalization of BIG4 (UBQ10::BIG4-YFP; **M**) and Golgi-marker γCOP (**N**) indicated no co-localization (**O**; **P**, intensity–line profile). (**I–K**, **M–O**) Scale bar, 5 µm.

The following figure supplements are available for figure 1:

**Figure supplement 1**. Expression and phylogeny of BIG ARF-GEFs.

**Figure supplement 2**. BIG3 and BIG4 localize at the TGN.

**Figure supplement 3**. Ultrastructural localization of BIG4-YFP and ultrastructural abnormalities in BFA-treated *big3* mutant seedling root cells.

*2010*) but not with Golgi marker COPI subunit γCOP (*Figure 1M–P*; *Movafeghi et al., 1999*). TGN localization of BIG4-YFP was confirmed by immunogold labeling on EM sections (*Figure 1—figure supplement 3A,B*). BIG3-YFP and BIG4-YFP co-localized with endocytic tracer FM4-64, labeling TGN after brief uptake (*Figure 1—figure supplement 2A–H*; *Ueda et al., 2001*; *Dettmer et al., 2006*). BIG3 and BIG4 also accumulated together with FM4-64 in BFA-induced post-Golgi membrane vesicle aggregates ('BFA compartments'), consistent with ultrastructural abnormalities in these aggregates and Golgi stacks in BFA-treated *big3* mutant (*Figure 1—figure supplement 2I–N, 3C–F*). Together, these data suggest a role for BIG1-4 in post-Golgi membrane trafficking.

## Secretory and vacuolar trafficking depend on BIG1 to BIG4 function

To identify trafficking routes regulated by BIG1-4, pathway-specific soluble and membrane-associated cargo proteins were analysed in BFA-treated wild-type and *big3* mutant seedlings (for a list of markers used, see *Supplementary file 1*; *Figure 2—figure supplement 1S,T*). Secretory GFP (secGFP) (*Viotti et al., 2010*), which is normally secreted from the cell, and plasma membrane (PM)-targeted syntaxin SYP132 were trapped in BFA compartments and did not reach the plasma membrane of *big3* seedlings, in contrast to wild-type, suggesting a role for BIG1-4 in late secretory traffic, that is from the TGN to the plasma membrane (*Figure 2A–D*). There was a slight retention of SYP132 in the BFA compartments of wild-type seedling roots, which probably reflects slowed-down passage of newly-synthesized proteins through the TGN. This becomes apparent upon BFA treatment because of TGN aggregation into BFA compartments, as has been reported earlier for *HS::secGFP* (*Viotti et al., 2010*). Vacuolar cargo proteins also pass through the TGN via multivesicular bodies (MVBs) to the vacuole (*Reyes et al., 2011*). Soluble RFP fused to phaseolin vacuolar sorting sequence AFVY accumulated in BFA compartments in *big3* mutant, in contrast to wild-type (*Scheuring et al., 2011*; *Figure 2E–J*, *Figure 2—figure supplement 1A–F*). Endocytosed PM proteins are delivered to the vacuole for degradation, for example boron transporter BOR1 in response to high external boron concentration (*Takano et al., 2005*; *Figure 2K–N*). BFA treatment prevented boron-induced trafficking of BOR1 to the vacuole in *big3* mutant, but not in wild-type (*Figure 2L,N*). BOR1 was rapidly turned over in the vacuole of wild-type, leaving no trace of GFP (*Figure 2L*). As expected, ARF-GEF BIG4 and its putative cargo BOR1 co-localized in BFA compartments (*Figure 2—figure supplement 1G–I*). Thus, BIG1-4 mediate both late secretory and vacuolar trafficking from the TGN.

## Recruitment of clathrin adaptor complex AP-1 to the TGN requires BIG1 to BIG4 function

ARF-GEFs activate ARF GTPases, resulting in recruitment of vesicular coat proteins to the respective endomembrane compartment, such as COPI complex to Golgi stacks or adaptor protein (AP) complexes to post-Golgi compartments (*Robinson, 2004*). Like BIG1-4, AP-1 complex subunit muB2-adaptin (AP1M2) localizes to SYP61-labeled TGN and is required for late secretory and vacuolar trafficking (*Park et al., 2013*; *Teh et al., 2013*; *Wang et al., 2013*; *Figure 2—figure supplement 1P–R*).

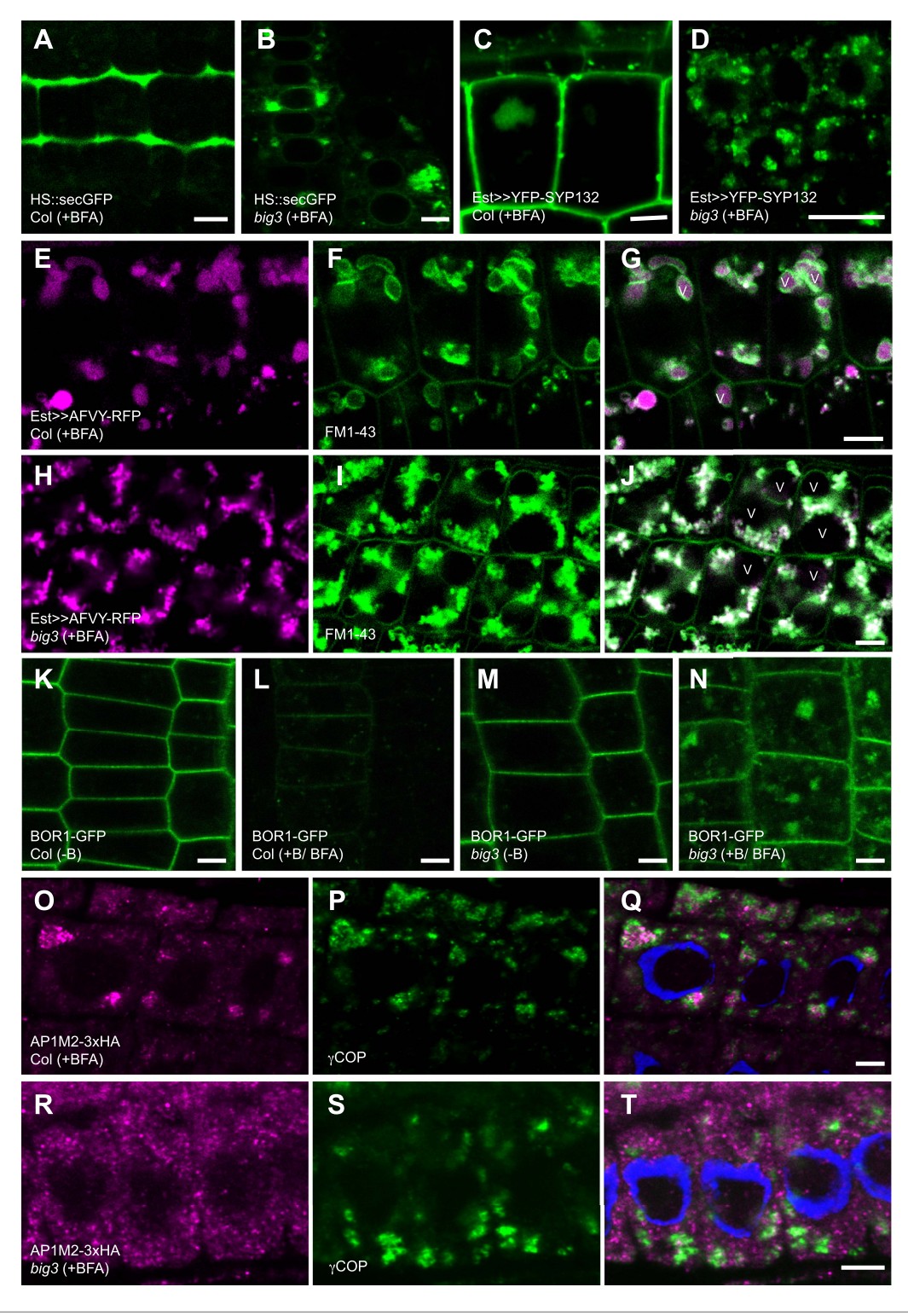

**Figure 2**. BIG1 – BIG4 regulate secretory and vacuolar trafficking by recruiting AP-1 adaptor complex. (**A** and **B**) BFA inhibited secretion of heat shock (HS)-induced secGFP in *big3* mutants (**B**) but not in wild-type (Col; **A**). (**C** and **D**) BFA inhibited trafficking of estradiol (Est)-induced YFP-SYP132 to the plasma membrane in *big3* mutants (**D**) but not in wild-type (Col; **C**). (**E–J**) BFA inhibited trafficking of soluble cargo AFVY-RFP to the vacuole (v), labeled by FM1-43 (**F** and **I**), in *big3* mutants (**H–J**) but not in wild-type (Col, **E–G**). (**K–N**) Live imaging of BOR1-GFP
*Figure 2. Continued on next page*

*Figure 2. Continued*

localization. Without boron (−B), BOR1-GFP localized at the plasma membrane in wild-type (**K**) and *big3* mutants (**M**). After BFA and boron treatment (+B), BOR1-GFP was degraded in the vacuole of wild-type (**L**) but accumulated in BFA compartments of *big3* mutants (**N**). (**O**–**T**) Immunostaining of 3xHA-tagged muB2 subunit of AP-1 complex (AP1M2; **O**, **R**) and COPI subunit γCOP (**P** and **S**) in BFA-treated seedlings. AP1M2 accumulated in BFA compartments surrounded by γCOP in wild-type (Col; **Q**). In *big3* mutants, γCOP was still recruited to Golgi membranes whereas AP1M2 was cytosolic (**R**–**T**). Blue, DAPI-stained nuclei. Scale bars, 5 μm.

The following figure supplements are available for figure 2:

**Figure supplement 1**. BIG1 – BIG4 regulate trafficking of secretory and vacuolar cargo by recruiting AP-1 complex.

---

AP1M2 also co-localized with TGN marker SYP61 in BFA compartments (*Figure 2—figure supplement 1J–L*). In BFA-treated *big3* mutant, however, AP1M2 was cytosolic whereas SYP61 was still TGN-associated (*Figure 2O,R*; *Figure 2—figure supplement 1J–O*). In contrast to AP1M2, Golgi association of COPI subunit γCOP, which is mediated by BFA-resistant ARF-GEF GNL1 (*Richter et al., 2007*), was not affected in BFA-treated *big3* mutant (*Figure 2O–T*). Thus, BIG1-4 specifically mediate AP-1 recruitment to the TGN.

## Secretion and recycling to the plasma membrane are independently regulated trafficking pathways

Another ARF-GEF in post-Golgi traffic, GNOM regulates polar recycling of auxin-efflux carrier PIN1 to the basal plasma membrane (*Geldner et al., 2003*). BFA treatment of wild-type and *big3* mutant seedlings inhibited recycling of PIN1, which accumulated in BFA compartments, and this defect was suppressed by engineered BFA-resistant GNOM (*Figure 3A–D*). Thus, BIG1-4 did not play any obvious role in PIN1 recycling. PIN1 is a stable protein such that most protein detectable at the plasma membrane is delivered via the recycling but not the secretory pathway (*Geldner et al., 2001*). In order to analyse the behavior of newly-synthesized PIN1 protein, we generated transgenic plants expressing estradiol-inducible PIN1. In contrast to recycling PIN1, newly-synthesized PIN1 protein was trapped in BFA compartments of *big3* mutant, regardless of BFA-resistant GNOM (*Figure 3E–H*). In conclusion, secretory ARF-GEFs BIG1-4 and recycling ARF-GEF GNOM regulate different post-Golgi trafficking pathways to the plasma membrane that function independently of each other.

Gravitropic growth response of the seedling root relies on GNOM-mediated PIN1 recycling (*Geldner et al., 2003*). We tested whether BIG1-4 are also required, using *DR5::NLS-3xGFP* expression to visualise auxin response (*Weijers et al., 2006*). BFA-induced inhibition of auxin response in wild-type and *big3* mutant was overcome by BFA-resistant GNOM, suggesting that BIG1-4 mediated secretion plays no role in gravitropic growth response (*Figure 4A–D*). GNOM-dependent PIN1 recycling is also required for lateral root initiation (*Geldner et al., 2003*). Surprisingly, BFA-resistant GNOM failed to initiate lateral root primordia in BFA-treated *big3* mutant in spite of stimulation by NAA, in contrast to seedlings that expressed both BIG3 and BFA-resistant GNOM (*Figure 4E–L*). *big3* mutants displayed binucleate cells, suggesting an essential role for secretory traffic in cytokinesis required for lateral root initiation (*Figure 4M–T*). For comparison, the BFA-induced defects in seed germination and primary root growth of *big3* were not rescued by engineered BFA-resistant GNOM, thus depending on secretory traffic rather than recycling (*Figure 1C,E,H*).

## Trafficking of both endocytosed and newly-synthesized proteins to the plane of cell division is regulated by secretory ARF-GEFs BIG1 to BIG4

In plant cytokinesis, which is assisted by a dynamic microtubule array named phragmoplast, both newly-synthesized and endocytosed proteins traffic to the plane of cell division on post-Golgi membrane vesicles that fuse with one another to form the partitioning cell plate (*Samuels et al., 1995*). This raises the problem of coordinating different trafficking routes in the brief period of mitotic division (*Reichardt et al., 2011*). Cell-plate formation requires cytokinesis-specific syntaxin KNOLLE, newly synthesized during late G2/M phase (*Lauber et al., 1997*; *Reichardt et al., 2007*). In contrast to wild-type, KNOLLE targeting to the division plane was inhibited in BFA-treated *big3* mutants, with KNOLLE accumulating in BFA compartments together with BIG4-YFP (*Figure 5A–F*, *Figure 5—figure supplement 1A–D*). Cell-plate formation was disrupted, resulting in binucleate cells, which sometimes

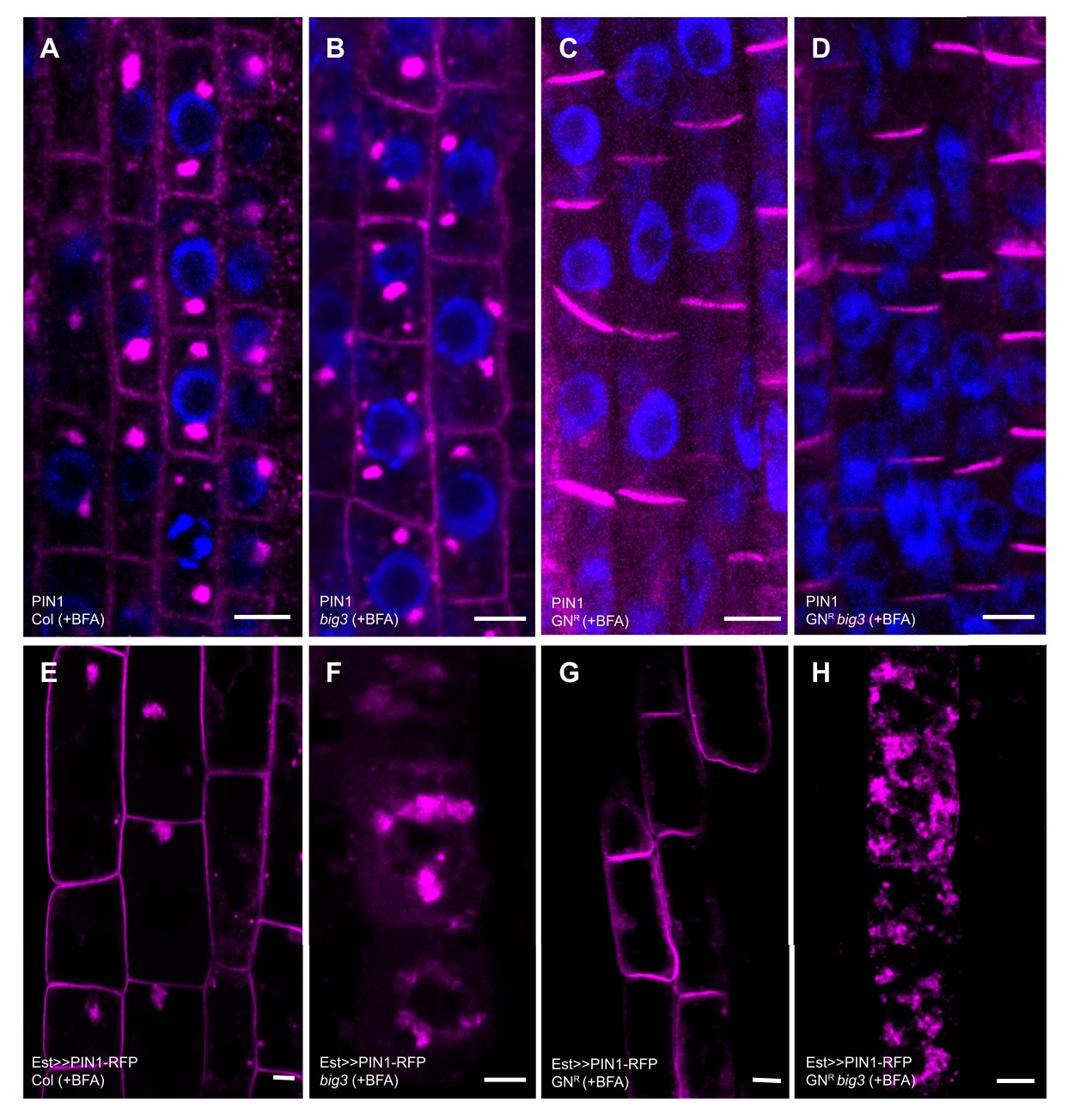

**Figure 3**. Secretion and recycling to the plasma membrane are regulated by different ARF-GEFs. (**A–D**) PIN1 localization in interphase cells of BFA-treated seedlings; apolar at the plasma membrane (PM) and in BFA compartments in wild-type (Col; **A**) and *big3* mutants (**B**); at the basal PM in BFA-resistant GN in wild-type (GN$^R$, **C**) or *big3* mutant background (GN$^R$ *big3*, **D**). Blue, DAPI-stained nuclei. (**E–H**) After BFA treatment, estradiol (Est)-induced PIN1-RFP was trafficked to the PM in wild-type (**E**) and BFA-resistant GN seedlings (GN$^R$, **G**) but not in *big3* mutants without (**F**) or with expression of BFA-resistant GN (GN$^R$ *big3*; **H**). Scale bars, 5 μm.

displayed cell-wall stubs (*Figure 5—figure supplement 2A–C*). We used the non-cycling plasma-membrane syntaxin SYP132 expressed from the strong mitosis-specific *KN* promoter as another secretory marker for trafficking to the cell–division plane (*Reichardt et al., 2011*). SYP132 also accumulated, together with KN, in BFA compartments of BFA-treated *big3* mutants, in contrast to BFA-treated wild-type (*Figure 5—figure supplement 1E–J*). We also analysed endocytosed plasma-membrane proteins PEN1 and PIN1 for BFA-sensitive trafficking to the cell plate in *big3* mutants. PEN1 syntaxin involved

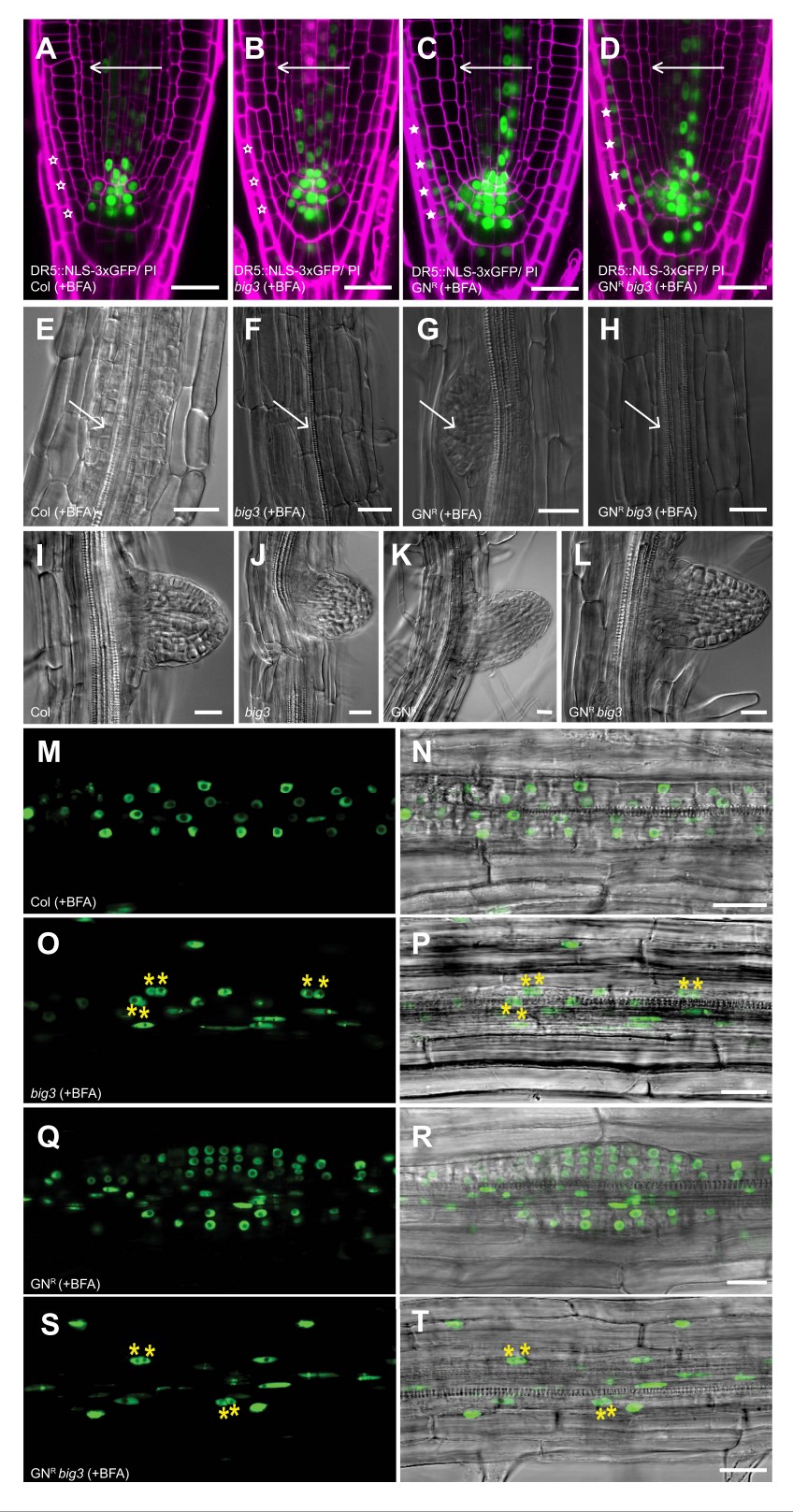

**Figure 4**. BIG1-4 in response to auxin application. (**A–D**) Visualization of auxin distribution by DR5::NLS-3xGFP (green) in BFA-treated seedlings after gravistimulation. Arrows, gravity vector. Cell walls were stained by propidium iodide (PI; magenta). Wild-type (**A**) and *big3* mutant seedling roots (**B**) did not respond to gravity (open asterisks),
*Figure 4. Continued on next page*

*Figure 4. Continued*
in contrast to BFA-resistant GN either in wild-type (GN$^R$, **C**) or *big3* mutant background (GN$^R$ *big3*, **D**). Asterisks, auxin response in epidermal cell layer on lower side (**C** and **D**). (**E–H**) NAA and BFA treatment led to proliferation of pericycle cells (arrows) in wild-type (**E**) but not *big3* mutants without (**F**) or with BFA-resistant GN (**H**). Normal lateral root primordia only formed in BFA-resistant GN (GN$^R$, **G**). Scale bars, 25 μm. (**I–L**) Bright-field microcopy of developing lateral root primordia in NAA-treated seedlings; genotypes: wild-type (Col; **I**), *big3* (**J**), BFA-resistant GN (GN$^R$; **K**) and BFA-resistant GN in *big3* mutant background (GN$^R$ *big3*; **L**). (**M–T**) Live imaging of DR5::NLS-3xGFP of seedling roots after NAA and BFA treatment. DR5::NLS-3xGFP signals (left panels **M**, **O**, **Q**, **S**) overlaid with Nomarski images (right panels **N**, **P**, **R**, **T**). Pericycle cells proliferated in wild-type (**M** and **N**) but became binucleate (asterisks) in *big3* (**O** and **P**) and GN$^R$ *big3* (**S** and **T**) mutants. Normal lateral root primordia were only formed in BFA-resistant GN (GN$^R$; **Q**, **R**) mutant. Scale bars, 25 μm.

in non-host immunity accumulates at the pathogen entry site by GNOM-dependent relocation following endocytosis from other regions of the plasma membrane (*Collins et al., 2003*; *Nielsen et al., 2012*). PEN1 continually cycles between plasma membrane and endosomes in interphase and accumulates at the cell plate in cytokinesis (*Reichardt et al., 2011*). To make sure that we were only looking at endocytosed PEN1, PEN1 was expressed from a histone H4 expression cassette that limits protein synthesis to S phase (*Reichardt et al., 2011*). In wild-type, BFA treatment inhibited PEN1 recycling to the plasma membrane but not its trafficking to the cell plate (*Reichardt et al., 2011*; *Figure 5G–I*). In contrast, in BFA-treated *big3* mutants, endocytosed PEN1 was not trafficked to the cell division plane but accumulated, together with KNOLLE, in BFA compartments (*Figure 5J–L*, asterisks). Endocytosed PIN1 trafficked, like KNOLLE, to the cell plate in BFA-treated wild-type but both PIN1 and KNOLLE were trapped in BFA compartments of *big3* mutants (*Figure 5M–R*). Expression of engineered BFA-resistant GNOM did not overcome the trafficking block to the division plane but rather diverted PIN1 to the basal plasma membrane (*Figure 5S–X*; compare *Figure 5X* with *Figure 5R*). Careful analysis of mitotic cells revealed polar accumulation of PIN1 at the plasma membrane of BFA-resistant GNOM seedling roots throughout mitosis while additional PIN1 accumulates at the forming and expanding cell plate, suggesting that trafficking to the plane of division and polar recycling to the plasma membrane occur simultaneously (*Figure 5—figure supplement 3*). Thus, both endocytosed and newly-synthesized plasma-membrane proteins require secretory ARF-GEF function BIG1-4 for trafficking to the plane of cell division.

## Discussion

It is a particularity of Arabidopsis and some other flowering-plant species that the secretory pathway of membrane traffic is comparatively insensitive to BFA treatment whereas endosomal recycling of endocytosed plasma-membrane proteins is rather sensitive (*Geldner et al., 2001*, *2003*; *Teh and Moore, 2007*; *Richter et al., 2007*). The BFA insensitivity of the secretory pathway depends on the BFA resistance of ARF-GEF GNL1, which mediates COPI-vesicle formation in retrograde Golgi-ER traffic (*Teh and Moore, 2007*; *Richter et al., 2007*), and also requires another BFA-resistant ARF-GEF acting in post-Golgi traffic to the plasma membrane. Here we show that ARF-GEFs BIG1-4 act at the TGN to mediate secretion of newly synthesized proteins to the plasma membrane in interphase but not recycling of endocytosed plasma-membrane proteins, and that BIG3 is BFA-resistant, unlike GNOM involved in recycling to the plasma membrane. Thus, there are two distinct trafficking pathways from the TGN to the plasma membrane in interphase. This is best illustrated by the trafficking of auxin-efflux carrier PIN1 - whereas newly synthesized PIN1 requires BIG1-4 on the late secretory pathway for non-polar delivery to the plasma membrane, polar PIN1 recycling to the basal plasma membrane solely depends on ARF-GEF GNOM (see model in *Figure 5—figure supplement 4*).

Like newly synthesized proteins, endocytosed proteins are targeted to the division plane during cytokinesis (*Reichardt et al., 2011*). Proteins that cycle between endosomes and the plasma membrane in interphase accumulate, preferentially or even exclusively, at the cell plate (*Reichardt et al., 2011*). In general, recycling to the plasma membrane appears to be switched off during cytokinesis. Here we show that secretory ARF-GEFs BIG1-4 are essential for protein trafficking to the plane of cell division, regardless of proteins being newly synthesized or endocytosed from the plasma membrane (see model in *Figure 5—figure supplement 4*).

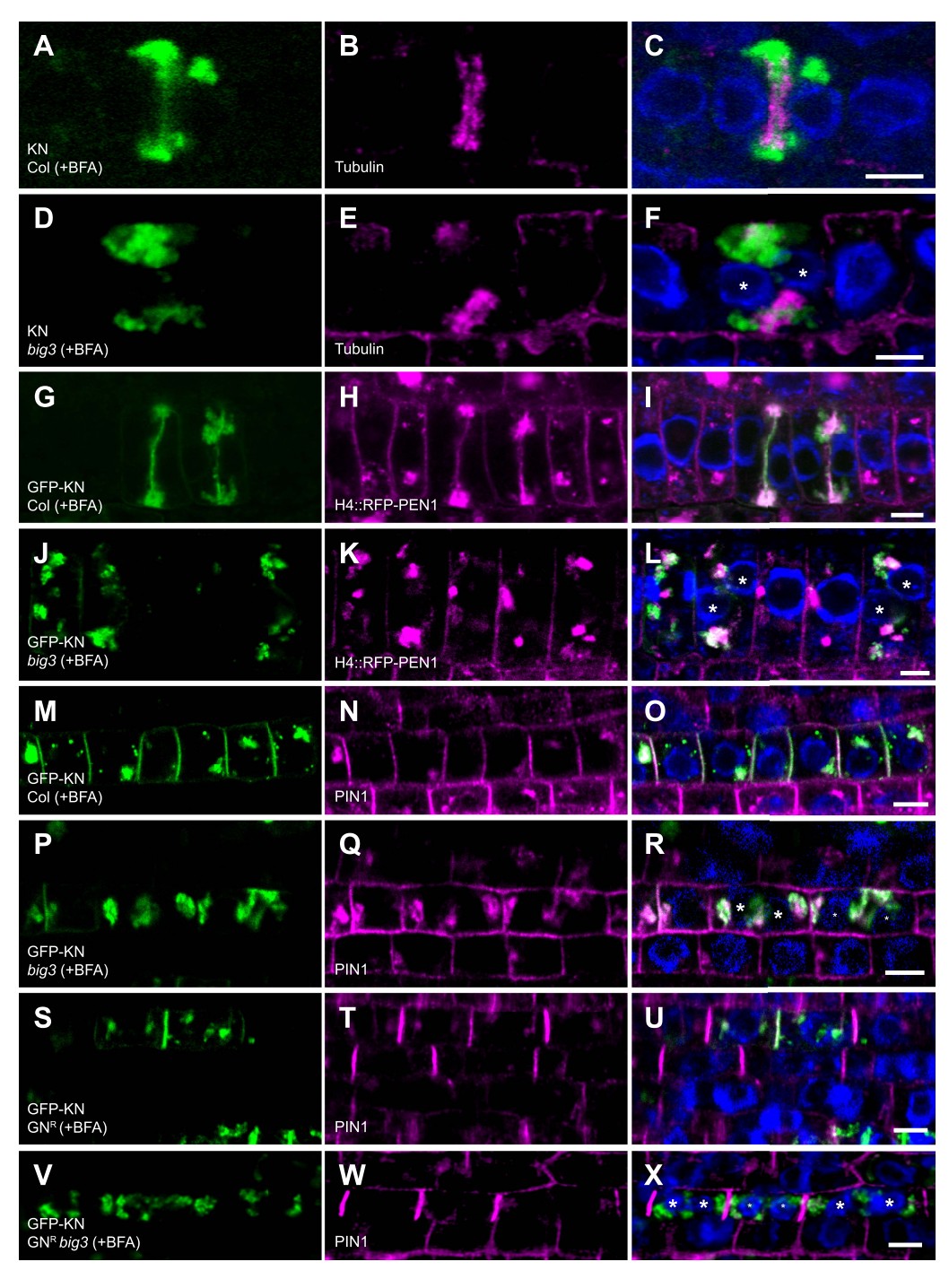

**Figure 5**. Trafficking to the plane of cell division is mediated by BIG1 – BIG4. (**A–F**) Immunolocalization of KNOLLE (KN; **A**, **D**) and tubulin (**B** and **E**) in cytokinetic root cells of BFA-treated seedlings (50 μM for 3 hr). (**A–C**) KN was located at the cell plate (**A**) flanked by tubulin-positive phragmoplast (**B**) in wild-type. (**D–F**) In *big3* mutants, KN accumulated in BFA compartments separated from tubulin-positive phragmoplast, resulting in a binucleate cell. (**G–L**) Co-localization of GFP-tagged KN and endocytosed RFP-PEN1 (H4::RFP-PEN1) in BFA-treated seedlings. KN and PEN1 co-localized at the cell plate and in BFA compartments of wild-type (**G–I**) but only in BFA compartments in *big3* mutants (**J–L**). (**M–X**) Immunostaining of GFP-KN and PIN1 in cytokinetic root cells of BFA-treated seedlings. (**M–R**) PIN1 localized apolarly at the plasma membrane (PM) and co-localized with KN in BFA compartments and at the cell plate in wild-type (**M–O**) but only in BFA-compartments in *big3* mutants (**P–R**). (**S–U**) In GNᴿ, PIN1 localized

*Figure 5. Continued on next page*

*Figure 5. Continued*

polarly at the plasma membrane (**T**) and co-localized with KN (**S**) at the cell plate (**U**). (**V**–**X**) Although PIN1 localized polarly at the PM (**W**) in GN^R *big3*, neither PIN1 (**W**) nor KN (**V**) was located at the cell plate. Blue, DAPI-stained nuclei. Asterisks label nuclei of binucleate cells (**F**, **L**, **R**, **X**). Scale bars, 5 µm.

The following figure supplements are available for figure 5:

**Figure supplement 1**. BIG4 and cargo proteins trapped in BFA compartments of dividing cells in BFA-treated *big3* mutant seedlings.

**Figure supplement 2**. Ultrastructural appearance of cryofixed, freeze-substituted and resin-embedded *big3* seedling root tips treated with BFA.

**Figure supplement 3**. PIN1 recycling in mitotic cells.

**Figure supplement 4**. Highly schematic model of secretory and recycling trafficking pathways in interphase and cytokinesis.

---

Although trafficking to the plane of cell division appears to override recycling of endocytosed proteins to the plasma membrane, we noticed one clear exception—auxin-efflux carrier PIN1, which accumulates polarly at the plasma membrane in interphase and during cell division when both BFA-resistant BIG3 and engineered BFA-resistant GNOM were expressed. Rather than substituting for BIG1-4 in traffic to the plane of cell division, recycling ARF-GEF GNOM appeared to counteract that process by promoting PIN1 recycling to the basal plasma membrane. Of course, the critical question is whether both processes occur at the same time or whether GNOM-dependent PIN1 recycling only sets in after trafficking to the cell plate has come to an end. Although there are no time-course studies, which would be difficult to perform because the process is very fast, detailed analysis of dividing cells at different mitotic stages revealed that polar recycling mediated by BFA-resistant GNOM occurs throughout mitosis and cytokinesis. Furthermore, only in the absence of both BFA-resistant BIG3 and BFA-resistant GNOM is PIN1 trapped in BFA compartments. If then BFA-resistant GNOM is expressed PIN1 is not delivered to the plane of division but rather polarly recycled to the plasma membrane, again suggesting that the latter pathway is a direct route bypassing the cell plate. PIN1 might be exceptional because continuous recycling of PIN1 is required for maintaining the polar transport of auxin across tissues (*Geldner et al., 2003*). If PIN1 recycling were shut down during cytokinesis this would disrupt the polar auxin transport required in specific developmental situations such as forming lateral root primordia when essentially all cells proliferate (*Geldner et al., 2004*). Another problem in auxin flow arises from cell division when the partitioning membrane has physically separated the two daughter cells: one daughter suddenly has PIN1 located at opposite ends. Obviously, PIN1 has to be removed from the wrong end in order to sustain polar auxin transport. This seems to be a fast process and has been studied for the related auxin-efflux carrier PIN2 in detail (*Men et al., 2008*).

Animal and plant cytokinesis differ in the way the partitioning membrane is laid down. In animals, secretory and recycling pathways contribute to the ingrowth of the plasma membrane mediated by a contractile actomyosin ring and to the subsequent abscission of the daughter cells (*Schiel and Prekeris, 2013*). In plants, a massive flow of membrane vesicles from TGN/early endosome to the plane of cell division sustains, by fusion, the rapid formation and outward expansion of the partitioning cell plate (*Samuels et al., 1995*). This process is orchestrated by a specialised cytoskeletal array termed phragmoplast that delivers those membrane vesicles to the division plane. Phragmoplast-assisted trafficking might be required for completing the partitioning membrane on time, in the absence of a cytokinesis-interphase checkpoint, and would thus effectively rule out recycling of endocytosed proteins to the plasma membrane. However, our results make clear that this is not the case because recycling to the plasma membrane is not switched off during cytokinesis. Rather, endocytosed proteins enter the late-secretory pathway to reach the division plane at the expense of being recycled to the plasma membrane, which requires the late-secretory ARF-GEFs BIG1-4. In conclusion, our results raise the possibility that in general, different ARF-GEFs have different specificity of action during vesicle formation such that the same cargo protein can be delivered to different destinations.

# Materials and methods

## Plant material and growth conditions

Plants were grown on soil or agar plates in growth chambers under continuous light conditions at 23°C. *big* mutant lines: *big1* (GK-452B06) and *big2* (GK-074F08) T-DNA lines were from GABI-KAT (http://www.gabi-kat.de), *big3* (SALK_044617) and *big4* (SALK_069870) T-DNA lines from the SALK collection (http://signal.salk.edu/cgi-bin/tdnaexpress). *big3* mutant lines were selected on MS plates using kanamycin.

The following transgenic marker lines were used: H4::RFP-PEN1 (*Reichardt et al., 2011*) (expressed from *HISTONE4* (*H4*) promoter during S phase), KN::Myc-SYP132 (*Reichardt et al., 2011*) (expressed during lateG2/M phase), HS::secGFP (*Viotti et al., 2010*) (expressed from heat shock promoter), GFP-KN (*Reichardt et al., 2007*), BOR1-GFP (*Takano et al., 2005*), DR5::NLS-3xGFP (*Weijers et al., 2006*), VHA-a1-RFP (*Viotti et al., 2010*), AP1M2-3xHA (*Park et al., 2013*).

## T-DNA genotyping of *big* mutant lines

Primers used to test for *big1* heterozygosity:
5'GCAAGATCAGGGAAGACG 3' and 5'ACCAGAGGAAGGTGCTTCTTC 3'
Primers used to test for *big1* homozygosity:
5'TCGTCCCATCTTCTTCATTTG 3' and 5'ACCAGAGGAAGGTGCTTCTTC 3
Primers used to test for *big2* heterozygosity:
5'GCAAGATCAGGGAAGACG 3' and 5'TTGAGGGGTTCATATGACAGC 3'
Primers used to test for *big2* homozygosity:
5'TTTCCCACTTTTTCCACTGTG 3' and 5'TTGAGGGGTTCATATGACAGC 3'
Primers used to test for *big3* heterozygosity:
5'AAACTCTCCACTGGCTAAGCC 3' and 5'ATTTTGCCGATTTCGGAAC 3'
Primers used to test for *big3* homozygosity:
5'AAACTCTCCACTGGCTAAGCC 3' and 5'GCAAGTTTTCTTGCGCAATAC 3'
Primers used to test for *big4* heterozygosity:
5'ATTTTGCCGATTTCGGAAC 3' and 5'CTATCTTGCGCTGGAGACAAC 3'
Primers used to test for *big4* homozygosity:
5'TCCTCTTCAAACTCGTCAACG 3' and 5'CTATCTTGCGCTGGAGACAAC 3'

## Generating transgenic plants

Genomic BIG4 was amplified and introduced into pDONR221 (Invitrogen, Darmstadt, Germany) and afterwards into *UBQ10::YFP* destination vector (*Grefen et al., 2010*). For generation of BFA-resistant *UBQ10::BIG4$^R$-YFP*, methionine at position 695 was exchanged with leucine by site-directed mutagenesis. *BIG3* promoter was amplified and introduced into pUC57L4 via *Kpn*I and *Sma*I restriction sites. Multistep gateway cloning was performed using pUC57L4-*BIG3*-promoter, pEntry221-*BIG4* and R4pGWB553 (*Nakagawa et al., 2008*) yielding *BIG3::BIG4-RFP*. Cloning the CDS from *BIG3* into pGREENII via *Apa*I and *Sma*I restriction sites generated pGII-*BIG3*. The 1 kb *BIG3* promoter was amplified and introduced into pGII-*BIG3* via *Apa*I. 1 kb of 3'UTR was amplified and introduced into pGII-*BIG3::BIG3* via *Sma*I and *Spe*I. C-terminal YFP was inserted via *Sma*I and *Spe*I. *AFVY-RFP* was amplified from *35S::AFVY-RFP* (*Scheuring et al., 2011*) and introduced into pDONR221 (Invitrogen) generating a pEntry clone. Afterwards, LR reaction was performed introducing *AFVY-RFP* into the estradiol-inducible destination vector pMDC7 (*Curtis and Grossniklaus, 2003*). *PIN1* cDNA was cloned into pGem-T (Promega, Mannheim, Germany). *RFP* was inserted in *PIN1* via the *Xho*I site. *PIN1-RFP* was amplified and introduced first into pDONR221 and then into pMDC7. *YFP-SYP132* was amplified and introduced into pDONR221 and then into pMDC7.

All constructs were transformed into *big3* mutants and BFA-resistant GN (GN$^R$) in *big3* mutant background. T1 plants of *UBQ10::BIG4-YFP*, *UBQ10::BIG4$^R$-YFP* and *BIG3-YFP* were selected by spraying with Basta. T1 seeds of estradiol-inducible lines and *BIG3::BIG4-RFP* were selected with hygromycin. Experiments were performed using T2 or T3 seedlings. At least three independent lines were analysed.

## Immunofluorescence localization and live imaging in seedling roots

5 days old seedlings were incubated in 1 ml liquid growth medium (0.5x MS medium, 1% sucrose, pH 5.8) containing 50 µM BFA (Invitrogen, Molecular Probes) for 1 hr or 3 hr at room temperature in 24-well

cell-culture plates. Seedlings treated with 50 µM BFA for (a) 1 hr or (b) 3 hr, respectively, were used for the following immunolocalisation studies: (a) AP1M2 vs γCOP, AP1M2 vs SYP61, PIN1; (b) KNOLLE vs Tubulin, KNOLLE vs PIN1, H4::RFP-PEN1 vs GFP-KN and KN::Myc-SYP132 vs KN. Incubation was stopped by fixation with 4% paraformaldehyde in MTSB. Immunofluorescence staining was performed as described (*Lauber et al., 1997*) or with an InsituPro machine (Intavis, Cologne, Germany) (*Müller et al., 1998*).

Antibodies used: mouse anti-MYC (Santa Cruz Biotechnology, Heidelberg, Germany) 1:600, mouse anti-HA 1:1000 (BAbCO, Richmond, CA, USA), rat anti-tubulin 1:600 (Abcam, Cambridge, UK), rabbit anti-PIN1 1:1000 (*Geldner et al., 2001*), rabbit anti-γCOP 1:1000 (Agrisera, Vännäs, Sweden), rabbit anti-KNOLLE 1:2000 (*Reichardt et al., 2007*) and rabbit anti-SYP61 1:700 (*Park et al., 2013*). Alexa-488 or Cy3-conjugated secondary antibodies (Dianova, Hamburg, Germany) were diluted 1:600.

Live-cell imaging was performed with 2 µM FM4-64 or FM1-43 (Invitrogen, Molecular Probes) or propidium iodide (10 µg/ml).

Estradiol induction was performed using 10 or 20 µM estradiol. BFA incubation (25 µM) was done together with estradiol for 6 hr.

Heat-shock inducible secGFP (HS::secGFP) lines were first incubated for 30 min at 37°C in MS at pH8.1. BFA treatment (50 µM) in MS at pH8.1 followed for 4 hr at plant room conditions.

Analysis of BOR1 degradation was performed according to *Takano et al. (2005)* . In addition, we treated the seedlings with BFA, 5 µM, for 1 hr together with boron.

## Electron microscopy

For ultrastructural analysis, root tips were high-pressure frozen (Bal-Tec HPM010; Balzers) in hexadecene (Merck Sharp and Dohme, Haar, Germany), freeze-substituted in acetone containing 2.5% osmium tetroxide, washed at 0°C with acetone, and embedded in Epon. For immunogold labeling of ultrathin thawed cryosections, root tips were fixed with 8% formaldehyde (2 hr), embedded in gelatin, and infiltrated with 2.1 M sucrose in PBS as previously described (*Dettmer et al., 2006*). Thawed ultrathin sections were labeled with rabbit anti-GFP antibodies (1:300; Abcam) and silver-enhanced (HQ Silver, 8 min; Nanoprobes, Yaphank, NY, USA) goat anti-rabbit IgG coupled to Nanogold (no. 2004; Nanoprobes). Antibodies and markers were diluted in blocking buffer (PBS supplemented with 0.5% BSA and 1% milk powder).

## Acquisition and processing of fluorescence images

Fluorescence images were acquired at 512 × 512 or 512 × 256 pixels with the confocal laser scanning microscope TCS-SP2 or TCS-SP8 from Leica, using the 63x water-immersion objective and Leica software. All images were processed with Adobe Photoshop CS3 only for adjustment of contrast and brightness. Intensity line profile was performed with Leica software.

## Pollen germination

Pollen medium was prepared as described (*Boavida and McCormick, 2007*). Pollen germinated over night or for 5 hr before microscopic analysis.

## Physiological tests

To investigate primary root growth, 5–6 days old seedlings were transferred to plates with 10 µM BFA and analysed after 5–7 additional days using ImageJ. DR5::NLS-GFP expressing seedlings analysed for lateral root formation were treated with 5 µM NAA or 5 µM NAA plus 10 µM BFA over night. Roots were cleared according to *Geldner et al. (2004)*. Gravitropic response was investigated by transferring 5 days old seedlings, expressing DR5::NLS-GFP, to BFA plates (5 µM). Seedlings were grown vertically for 1 hr on BFA plates before rotated by 135° for 4 hr.

For analysis of seed germination, seeds were sown out on MS medium containing 5 µM BFA. Images were taken after 5 days of growth.

## Phylogenetic tree

Full-length protein sequence of BIG3 was used to search for related sequences from different plant species with sequenced genomes that are available at the phytozome homepage (http://www.phytozome.net/). ARF-GEFs from different species were aligned by ClustalW (www.ebi.ac.uk/clustalw) and the phylogenetic tree was drawn with Dendroscope (*Huson et al., 2007*).

## Acknowledgements

We thank Lukas Sonnenberg and Marlene Ballbach for technical assistance, Toru Fujiwara, Niko Geldner, Christopher Grefen, Ueli Grossniklaus, Sumie Ishiguru, Peter Pimpl, Masao H. Sato and Karin Schumacher for sharing published materials, Joop Vermeer (Univ. Lausanne) for cloning vector pUC57L4, and NASC for T-DNA insertion lines. We also thank Martin Bayer, Niko Geldner, Christopher Grefen, Michael Hothorn and Steffen Lau for critical reading of the manuscript.

## Additional information

### Funding

| Funder | Grant reference number | Author |
|---|---|---|
| German Research Foundation (DFG) | SFB446/TP A9 | Gerd Jürgens |
| German Research Foundation (DFG) | JU 179/18-1 | Gerd Jürgens |
| Carlsberg Foundation | 2011_01_0789 | Mads Eggert Nielsen |

The funders had no role in study design, data collection and interpretation, or the decision to submit the work for publication.

### Author contributions

SR, Conception and design, Acquisition of data, Analysis and interpretation of data, Drafting or revising the article; MK, MEN, MP, RG, CK, UV, HB, UM, Y-DS, Acquisition of data, Analysis and interpretation of data; SB, Analysis and interpretation of data, Drafting or revising the article; GJ, Conception and design, Analysis and interpretation of data, Drafting or revising the article

## Additional files

### Supplementary files

• Supplementary file 1. Localization of vesicle trafficking markers. This table summarizes the localization of different vesicle trafficking markers without BFA (1th column) and with BFA in wild-type (Col; 2th column), *big3* (3th column), BFA-resistant GNOM (GN^R; 4th column) and BFA-resistant GNOM in *big3* mutant background (GN^R *big3*; 5th column). Abbreviations: PM, plasma membrane; CP, cell plate; BFA-comp., BFA-compartment.

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
