## [Decision Letter]

Thank you for sending your work entitled “Recycling-to-secretory ARF-GEF switch mediating protein traffic to the cell division plane” for consideration at *eLife*. Your article has been favorably evaluated by a Senior editor, a Reviewing editor, and 3 reviewers.

The Reviewing editor and the reviewers discussed their comments before reaching this decision, and the Reviewing editor has assembled the following comments to help you prepare a revised submission.

All three reviewers agree that your paper addresses important unresolved issues and that both experiments and data presentation are of exceptional quality. There is however some discussion regarding your conclusion that “endocytosed proteins enter the late secretory pathway to reach the division plane at the expense of being recycled to the plasma membrane, which is regulated by the switch to the late secretory ARF-GEFS BIG1-4” that we would like you to address. The reviewers consider this is a significant claim because the identity of the cell-plate pathway and its relationship to pathways that operate in interphase has been a longstanding unresolved controversy. However, they fail to see how these specific conclusions can be drawn from the present data. We would like to ask you to consider the following reviewer comments in detail:

1) Although the data show that traffic of all cargoes to the cell plate requires BIG1-4, I see no evidence that BIG1-4 REGULATE a switch of cargoes from one pathway to another.

2) Second, an important piece of evidence is that in BFA-treated *big3* GN^R^ plants PIN1 proteins recycle to their normal polar position at the basal PM during cytokinesis (Figure 5). This implies that recycling continues during cytokinesis so traffic to the cell plate requires sorting of PIN1 away from that pathway into the BIG1-4 pathway. I think the authors are using the presence of the cytokinesis-specific syntaxin KNOLLE to identify cells undergoing cytokinesis. Normally this is reasonable because KNOLLE is so rapidly degraded in the vacuole after cytokinesis but I am concerned here that in BFA-treated *big3* plants KNOLLE may be artificially stabilised as trafficking to the vacuole is inhibited under these conditions (Figure 2). Indeed the extensive labelling of whole files of cells is unusual for a marker of cytokinesis suggesting that there may be temporal stability. Treatment seems to have been for 3 h so the majority of cells that contain KNOLLE may have gone through (failed) cytokinesis some time ago. Isn't it possible that recycling to the plasma membrane does not in fact occur during cytokinesis, that endocytosed PIN1 accumulated in BFA-compartments during cytokinesis, but was recycled back to the PM in the subsequent G1. If so the important change could be the inhibition of the GN-pathway during cytokinesis causing endocytosed proteins to enter the default secretory pathway at the TGN, rather than any regulatory role for BIGs.

3) Even if recycling to the PM does occur during cytokinesis, I do not think the data shows compellingly that there is indeed such a switch (during cell plate assembly) from the GN-dependent recycling pathway to the BIG1-4-dependent secretory pathway.

a) Although the data show that GNOM is sufficient to recycle a majority of endocytosed protein, can it be excluded that a fraction of endocytosed protein that enters the TGN in interphase is returned by the BIG1-4 secretory pathway? If so, it could be this fraction that arrives in the cell plate during cytokinesis. This might explain why endocytosed PIN1 in BFA-treated wild-type appears to accumulate in BFA-compartments (cell plate associated and detached) as well as the cell plate.

b) Although some endocytosed protein arrives in the cell plate, is it clear that all or most endocytosed protein goes there as proposed?

4) To resolve question 2 convincingly I think the authors will need at least to show time-lapse data on cells of relevant genotypes undergoing cytokinesis with and without BFA treatments. There probably needs to be an independent marker of cell cycle stage such as tubulin-GFP, which would also provide evidence that phragmoplasts are normal in the BFA-treated cells. I also think that it would be important to ask whether newly-synthesised PIN1 is trapped in interphase BFA compartments in *big3* and *big3* GN^R^. To resolve the second question, perhaps photoswitchable fluorescent proteins which are available may also be suitable though I realise that they are not easy to image. Protein could be photoconverted in the BFA compartment of BFA-treated Col and followed over time to look at its rate of return to the PM versus the cell plate when cells enter cytokinesis – I don't recall this specific experiment being been done before.

5) From our perspective, the time-lapse (or equivalent) data would be necessary to establish the crucial point about recycling during cytokinesis. We imagine you should be in a position to do this without much delay and to establish the stability of KN in *big3*+BFA - one reviewer suggests that the conditionality of the BFA-induced phenotype is a perfect system for such experiments. The photoswitch experiment or similar is much more demanding but without it there is ambiguity about whether anything really changes with respect to PIN1 trafficking during cytokinesis, a major claim of the paper. If this point remains unclarified, we think the Discussion and title will need to change to reflect this.

6) The reviewers also question the coining of BIG1-4 as “regulatory switches”. It seems that the paper does not demonstrate a direct molecular mechanism in which BIG1-4 display a switch-like function, rather you seem to provide genetic evidence that BIG1-4 are necessary for the observed changes in trafficking. At this point, alternative hypotheses, e.g., that GN or something else on the GN pathway is somehow inactivated during cytokinesis, also appear plausible. We understand that this is partly a semantic issue, but we still would like to ask you to re-word appropriately.

We hope you will be able to address the above issue, but let us clarify that the remaining points that cell plate assembly requires BIG1-4 and so is derived from a biosynthetic pathway at the TGN, and that there are two distinct classes of TGN with distinct ARF-GEFs that keep biosynthetic and recycling cargo separate, are valuable contributions. For these alone, your paper merits publication, but it would have to be revised accordingly in the Discussion and title.

---

## [Author Response]

*1) Although the data show that traffic of all cargoes to the cell plate requires BIG1-4, I see no evidence that BIG1-4 REGULATE a switch of cargoes from one pathway to another*.

This is partly a semantic issue but we understand the concern and have toned down the statement, avoiding “regulation” and “switch”. In our view, nonetheless, BIG1-4 are absolutely required for trafficking to the cell plate. They cannot be substituted by e.g., GNOM, which rather mediates recycling to the plasma membrane (see also reply to the next comment). Thus, there is a specific trafficking pathway to the cell plate mediated by secretory ARF-GEFs and taken by both newly synthesised and endocytosed cargo proteins.

*2) Second, an important piece of evidence is that in BFA-treated* big3 *GN*^*R*^
*plants PIN1 proteins recycle to their normal polar position at the basal PM during cytokinesis (*Figure 5*). This implies that recycling continues during cytokinesis so traffic to the cell plate requires sorting of PIN1 away from that pathway into the BIG1-4 pathway. I think the authors are using the presence of the cytokinesis-specific syntaxin KNOLLE to identify cells undergoing cytokinesis. Normally this is reasonable because KNOLLE is so rapidly degraded in the vacuole after cytokinesis but I am concerned here that in BFA-treated* big3 *plants KNOLLE may be artificially stabilised as trafficking to the vacuole is inhibited under these conditions (*Figure 2*). Indeed the extensive labelling of whole files of cells is unusual for a marker of cytokinesis suggesting that there may be temporal stability. Treatment seems to have been for 3 h so the majority of cells that contain KNOLLE may have gone through (failed) cytokinesis some time ago*.

We agree that KNOLLE might not be a reliable marker in this situation. The experiment has been repeated, using anti-tubulin immunostaining of mitotic microtubule arrays and DAPI staining of nuclear chromatin as reliable markers for mitotic staging of dividing cells. The new results reveal polar localisation of PIN1 at the plasma membrane in dividing cells of GN-BFAr at successive stages of mitosis (metaphase to telophase) and cytokinesis (cell plate being formed, laterally expanded or nearly complete), this being accompanied by PIN1 accumulation at the cell plate during cytokinesis (see new Figure 5—figure supplement 3). The control images (wild-type, i.e., GNOM inactivated by BFA) also show PIN1 accumulation at the cell plate but no polar accumulation at the basal plasma membrane. These data suggest competitive traffic to both cell plate and basal plasma membrane during cytokinesis (see last paragraph of the Results, and Discussion).

*Isn't it possible that recycling to the plasma membrane does not in fact occur during cytokinesis, that endocytosed PIN1 accumulated in BFA-compartments during cytokinesis, but was recycled back to the PM in the subsequent G1. If so the important change could be the inhibition of the GN-pathway during cytokinesis causing endocytosed proteins to enter the default secretory pathway at the TGN, rather than any regulatory role for BIGs*.

See reply to preceding comment.

*3) Even if recycling to the PM does occur during cytokinesis, I do not think the data shows compellingly that there is indeed such a switch (during cell plate assembly) from the GN-dependent recycling pathway to the BIG1-4-dependent secretory pathway*.

We agree that the underlying mechanism is unknown. We have replaced “switch” by “change” from recycling to secretory trafficking pathway.

*a) Although the data show that GNOM is sufficient to recycle a majority of endocytosed protein, can it be excluded that a fraction of endocytosed protein that enters the TGN in interphase is returned by the BIG1-4 secretory*
*pathway*?

This is a rather theoretical possibility. It has been documented many times since the first analysis published in [7] that GNOM is sufficient for PIN1 recycling and here we show that BIG3 does not make any difference.

*If so, it could be this fraction that arrives in the cell plate during cytokinesis. This might explain why endocytosed PIN1 in BFA-treated wild-type appears to accumulate in BFA-compartments (cell plate associated and detached) as well as the cell plate*.

When BIG3 is deleted and GNOM rendered BFA-resistant PIN1 does not reach the cell plate but is properly recycled to the basal plasma membrane (see also reply to preceding comment).

*b) Although some endocytosed protein arrives in the cell plate, is it clear that all or most endocytosed protein goes there*
*as proposed*?

A previous quantitative study of endocytosed PEN1 distribution in cytokinesis cells indicated that most PEN1 accumulates at the cell plate with little, if any, signal remaining at the plasma membrane (23).

*4) To resolve question 2 convincingly I think the authors will need at least to show time-lapse data on cells of relevant genotypes undergoing cytokinesis with and without BFA treatments*.

This is technically challenging and beyond the scope of the present paper. However, we think that our detailed analysis of dividing cells at different stages of mitosis and cytokinesis essentially provides the relevant information. We have now used anti-tubulin immunostaining of mitotic microtubule arrays and DAPI staining of nuclear chromatin as reliable markers for mitotic staging of dividing cells. The new results reveal polar localisation of PIN1 at the plasma membrane in dividing cells of GN-BFAr at successive stages of mitosis (metaphase to telophase) and cytokinesis (cell plate being formed, laterally expanded or nearly complete), this being accompanied by PIN1 accumulation at the cell plate during cytokinesis (see new Figure 5—figure supplement 3). The control images (wild-type, i.e., GNOM inactivated by BFA) also show PIN1 accumulation at the cell plate but no polar accumulation at the basal plasma membrane. These data suggest competitive traffic to both cell plate and basal plasma membrane during cytokinesis (see last paragraph of Results, and Discussion).

*There probably needs to be an independent marker of cell cycle stage such as tubulin-GFP, which would also provide evidence that phragmoplasts are normal in the BFA-treated cells*.

We have used both the DAPI-visualised nuclear cycle and anti-tubulin staining of microtubule arrays (see new Figure 5—figure supplement 3).

*I also think that it would be important to ask whether newly-synthesised PIN1 is trapped in interphase BFA compartments in* big3 *and* big3 *GN*^*R*^.

That was done (Figure 3): PIN1 expressed from the estradiol-inducible promoter was trapped in *big3* and *big3* GN^R^.

*To resolve the second question, perhaps photoswitchable fluorescent proteins which are available may also be suitable though I realise that they are not easy to image. Protein could be photoconverted in the BFA compartment of BFA-treated Col and followed over time to look at its rate of return to the PM versus the cell plate when cells enter cytokinesis* – *I don't recall this specific experiment being been done before*.

As mentioned in our reply above, this is technically challenging and beyond the scope of the present paper. However, we have used both the DAPI-visualised nuclear cycle and anti-tubulin staining of microtubule arrays (see new Figure 5—figure supplement 3).

*5) From our perspective, the time-lapse (or equivalent) data would be necessary to establish the crucial point about recycling during cytokinesis. We imagine you should be in a position to do this without much delay and to establish the stability of KN in* big3 *+BFA*.

Just to summarise our replies here, we have repeated the experiment, using DAPI-visualised nuclear cycle and anti-tubulin staining of microtubule arrays for staging of dividing cells (see new Figure 5—figure supplement 3). Unlike KN, these markers are not likely affected by changes in trafficking conditions. We detected PIN1 polar accumulation at the basal plasma membrane at all mitotic stages from metaphase to anaphase to telophase as well as during initiation, expansion and completion of the cell plate, revealing progression of cytokinesis. During cytokinesis, polar accumulation at the PM was accompanied by accumulation at the cell plate when both BFA-resistant BIG3 was present and GNOM was rendered BFA-resistant.

*One reviewer suggests that the conditionality of the BFA-induced phenotype is a perfect system for such experiments*.

We agree, in principle. However, the dynamics of the process is enormous, and our previous experience with GFP-tagged KN suggests that the short duration of cytokinesis makes live-imaging challenging. Thus, the effort required may well be beyond the scope the present paper.

*The photoswitch experiment or similar is much more demanding but without it there is ambiguity about whether anything really changes with respect to PIN1 trafficking during cytokinesis, a major claim of the paper. If this point remains unclarified, we think the Discussion and Title will need to change to reflect this*.

See our new data on Col vs. GN-BFAr and the expanded Discussion. We have modified the Results and Discussion sections accordingly.

*6) The reviewers also question the coining of BIG1-4 as “regulatory switches”. It seems that the paper does not demonstrate a direct molecular mechanism in which BIG1-4 display a switch-like function, rather you seem to provide genetic evidence that BIG1-4 are necessary for the observed changes in trafficking. At this point, alternative hypotheses, e.g., that GN or something else on the GN pathway is somehow inactivated during cytokinesis, also appear plausible. We understand that this is partly a semantic issue, but we still would like to ask you to re-word appropriately*.

We agree that we cannot present a molecular mechanism regulating the pathway change of endocytosed cargo proteins in cytokinesis. However, our results strongly suggest that GNOM is still functional during mitotic and cytokinetic stages. This implies to us that the destination of cargo proteins, division plane or plasma membrane, depends on whether they are recruited to membrane vesicles whose formation is mediated by BIG1-4 or GNOM, respectively.

*We hope you will be able to address the above issue, but let us clarify that the remaining points that cell plate assembly requires BIG1-4 and so is derived from a biosynthetic pathway at the TGN, and that there are two distinct classes of TGN with distinct ARF-GEFs that keep biosynthetic and recycling cargo separate are valuable contributions. For these alone, your paper merits publication, but it would have to be revised accordingly in the Discussion and Title*.

In order to avoid confusion for the reader, we have changed the title and the Discussion as suggested and make no claim regarding switches or molecular regulation of the process.